# OpenReview forum: "Federated Learning under Evolving Distribution Shifts"
_TMLR — Rejected by TMLR_

### Review · Reviewer_UXUd · 2025-04-23

**Summary Of Contributions:**

This paper addresses federated learning (FL) under evolving distribution shifts, proposing two methods namely FedEvolve and FedEvp to capture evolving patterns in client data and generalize to unseen future domains. FedEvolve learns evolving patterns explicitly via dual representation mappings for consecutive domains, while FedEvp simplifies this by using a single evolving-domain-invariant representation mapping with personalization to reduce overhead. The authors empirically demonstrate significant performance improvements over various FL baselines on multiple synthetic and real datasets.

**Audience:**

No

**Claims And Evidence:**

No

**Requested Changes:**

## **Critical**:

- Clearly identify and describe specific real-world scenarios where the FedEvolve and FedEvp methods offer practical and meaningful advantages over existing approaches, particularly in situations where all three key challenges of privacy, heterogeneity, and evolving distribution shifts are present.

- Clearly state the high-level idea behind FedEvolve and FedEvp in the abstract.

- Provide empirical evidence to support the claim that existing FL methods fail to converge or perform poorly under evolving distribution scenarios. If such evidence is not available, consider softening the claim accordingly.

- Address limitations explicitly in the discussion, particularly conditions under which FedEvp and FedEvolve may underperform.

- In datasets where the distribution evolution is relatively simple (e.g., Rotated MNIST, Rotated EMNIST, Circles), learning the mapping between prototypes across time steps is straightforward. As a result, the proposed method demonstrates significant improvement over prior work. However, in more realistic datasets where distribution shifts are more semantic in nature (e.g., Portraits and Caltran), the improvement is less pronounced or even not exist, particularly on the server side. This raises concerns about the scalability of the method to real-world scenarios involving complex, high-level distribution shifts.

-  The authors claim that most prior works 'still assume that the data distribution of each client is static and, in particular, remains fixed between training and testing.' However, this statement is not entirely accurate, as federated domain generalization specifically addresses scenarios where the training and testing distributions differ.

- Although Table 4 provides results with a moderately non-iid (Dir=1) setting, to support the claim that your model effectively addresses heterogeneity, experiments with smaller Dir values are expected, following the established values for Dir of heterogeneity in prior works [1].

## **Strengthening**:

- Introduce and define acronyms clearly and consistently, preferably in uppercase.

- Consider including a variance-based measure (e.g., KL divergence assuming Gaussian distributions around prototypes) to capture shifts beyond mean differences.

- Elaborate on the results in Table 4 and Figure 3 regarding the observed gap between generalization and personalization as client heterogeneity (Dir) decreases.

- Clarify the redundancy between Tables 5 and 9 or justify why both are presented.

- Include a "Future Work" section highlighting avenues for further exploration, such as handling completely unpredictable shifts or improving computational efficiency.

- Provide sensitivity analysis concerning different distance metrics, as this choice is central to the method's design.


## **References**:

[1] Mendieta, Matias, et al. "Local learning matters: Rethinking data heterogeneity in federated learning." CVPR. 2022.

**Strengths And Weaknesses:**

## **Strengths**:

- The paper is organized properly and it is easy to follow.

- Comprehensive experimental evaluation across multiple datasets (Circle, RMNIST, EMNIST, Portraits, Caltran) and architectures (MLP, CNN, ResNet) demonstrates strong empirical evidence for the proposed methods.

- Theoretical motivation and bounds provided in the methodology help justify the design choices and offer insights into method robustness.

- The simplification from FedEvolve to FedEvp effectively addresses computational and communication overhead, making the approach practically appealing.

- Performance robustness against client heterogeneity and varying levels of evolving distribution shifts is mostly well-documented.

## **Weaknesses**:

- The significance of the proposed problme in real-word scenario is not well-justified.

- The abstract lacks a clear, high-level explanation of the proposed methods.

- Some claims in the motivation (e.g., "fail to converge") lack empirical support.

- The approach implicitly assumes a structured, predictable evolution pattern, which is simpler than fully unpredictable distribution shifts (as in Federated Domain Generalization - FDG).

- The method focuses on aligning prototypes by mean values without explicitly accounting for distribution variance, potentially overlooking important variance-based shifts.

- The paper lacks discussion of method limitations, particularly the boundaries of effectiveness or scenarios where performance could degrade significantly.

- In more realistic scenarios, the improvement over existing methods is marginal.

- The paper would benefit from a discussion of potential future research directions to guide subsequent work in this area.

---

> ### Author Response · Authors · 2025-06-26
> **Response to Reviewer UXUd**
>
> We are grateful for your insightful comments and suggestions. Below, we respond to each of your concerns point by point. Revisions made in the manuscript based on your suggestions are highlighted in **orange**.
>
> ## Weakness 1 and Requested Change 1: The significance of the proposed problem
>
> We included several real-world examples in the Introduction to better illustrate the relevance of our problem. These include satellite data with temporal environmental shifts and spatial heterogeneity, and clinical datasets where disease prevalence changes over time and heterogeneity arises due to differences in hospital infrastructure. Additionally, as suggested, the evolution of human language across time and regions serves as another practical example that we have updated into our manuscript. In all these scenarios, data distributions change dynamically over time and vary across geographic or institutional boundaries, leading to both heterogeneity and evolving distribution shifts. This is especially critical in medical settings, where strict privacy constraints prohibit the sharing of user data. Under such conditions, our proposed methods FedEvolve and FedEvp provide effective solutions for handling both distribution shifts and client heterogeneity without requiring data or prototype sharing. We have improved the problem statement to build a clearer connection between the abstract formulation and its real-world applications.
>
> ## Weakness 2 and Requested Change 2: High-level summary in abstract
>
> Thanks for your suggestions. We have revised the abstract to introduce FedEvolve and FedEvp in a clearer, method-focused manner. FedEvolve explicitly models the temporal evolution by learning two distinct representation mappings that capture the transition between consecutive data domains for each client. And FedEvp learns a single, evolving-domain-invariant representation by aligning current data with prototypes that are continuously updated from all previously seen domains.
>
> ## Weakness 3 and Requested Change 3: Some claims in the motivation (e.g., "fail to converge") lack empirical support.
>
> We have changed this saying to "show degradation under evolving shifts". We also added a short empirical note referencing Figure 3 that shows FL methods suffer performance drops under distribution shift to support the claim.
>
> ## Weakness 4: Structured evolution assumption
>
> In theoretical analysis, we assume the evolution pattern is structured to simplify the analysis. However, the real-world datasets do not follow the structured evolving patterns. For example, the face changing in the Portrait dataset and the light changing in Caltran are natural and not subject to any structured evolution.  We have updated the manuscript to acknowledge our assumption in the Limitation section and explicitly state that our methods target scenarios. We position this as complementary to Federated Domain Generalization.
>
> ## Weakness 5: Accounting for distribution variance
>
> Our approaches focus on aligning the first moment (mean) of the class-conditional feature distributions, even with first moment information, the proposed algorithms already show performance improvement. This may provide a motivation for developing algorithms that account for distribution variance, and is an interesting direction for future research. We acknowledge that accounting for distribution variance might improve the methods. A potential method is to use Mahalanobis distance, which also considers the covariance information. We have added this in the limitation section.
>
> ## Weaknesses 6-8 and Requested Change 4
>
> In the updated Appendix section F *Limitation*, we have highlighted potential failure cases that include highly nonstationary or chaotic domain shifts, other potential drifts like label shift and concept shift. We also add a discussion on conditions where FedEvp and FedEvolve may underperform and potential future works in this section.
>
> ## Requested Change 5: Scalability to complex real-world shifts
>
> On Portraits and Caltran, the improvements of our methods are more modest compared with synthetic datasets. This is likely due to more semantic and subtle distribution shifts (e.g., lighting, fashion), which are harder to capture via prototype alignment. It suggests that in such settings, domain evolution may not be sufficiently structured, or the learned representations may lack the granularity to model semantic variance. However, we still outperform existing methods even when the patterns are harder to capture. In addition, we have included the Reddit dataset (a text classification dataset) in our experiments, presented in Appendix E.2. We use the Reddit post titles from different years as domains to validate the effectiveness of our methods on language evolving changes.

---

> > ### Author Response · Authors · 2025-06-26
> >
> > ## Requested Change 7: Experiments with smaller Dir values are expected
> >
> > We have provided results with Dir=0.1 in revision Appendix E.6, and cited Mendieta et al. as suggested.
> >
> > ## Strengthening  1-3
> >
> > We have improved our manuscript based on your suggestions. We consider the variance-based measures as future works and introduce them in Appendix F. We have also extended the \textit{Impacts of distribution shifts and local heterogeneity} in Section 5.3 to better elaborate on results.
> >
> > ## Strengthening  4
> >
> > Thanks for reminder. We have removed the redundant table 9.
> >
> > ## Strengthening  5
> >
> > We have added a limitation and future work section at the end of Appendix.
> >
> > ## Strengthening  6
> >
> > We have included a new table to discuss the choice of distance metric, including cosine distance and L2 distance in Appendix E.3.

---

### Review · Reviewer_4V5m · 2025-04-27

**Summary Of Contributions:**

The paper presents two algorithms, FedEvolve and FedEvp, to improve Federated Learning in the presence of dynamic client data distributions. FedEvolve uses twice as many parameters and performs better, whereas FedEvp trades off performance for efficiency. Empirical results show strong performance for these algorithms against several baselines.

**Audience:**

Yes

**Broader Impact Concerns:**

Not applicable.

**Claims And Evidence:**

No

**Requested Changes:**

In addition to addressing the weaknesses mentioned above:
- A simplified version of the algorithm should be in the main text. The authors can move some things, like the impact of stragglers, to the appendix to make space.
- A figure with a Neural Network as an example demonstrating the parameters $\phi$ and $\psi$, and the algorithm would make it easier to understand.
- Add local-only baseline to show the impact of collaboration. This is especially important when the test set is not global and follows the data distribution of the training set.
- Can the authors explain why sometimes the average accuracy of clients' models is better than the accuracy of the averaged model (Circle, Caltran), and sometimes the trend is inverted (Portraits)?

**Strengths And Weaknesses:**

# Strengths
- Important research problem.
- Empirical comparison done against a wide range of baselines.

# Weaknesses
- The paper is very dense and difficult to follow. The following are a few details missing that make it difficult to understand what is going on:
  - Algorithm 1 has parameter *d* in line 16 and function *f* in line 12, which are not defined in the algorithm.
  - Algorithm 2 has parameter $\psi$ defined but *w* is used instead.
  - The number of clients is only mentioned in the appendix. This is an important detail that should be mentioned in the main text.
  - The composition of the test set is not described. Does the test set follow the same label distribution as the training set? If so, why? In the experiments, how many domains were used during training and how many for testing? Consequently, it is unclear from the paper what the test accuracy really means.
  - It is not clear from the paper whether all clients following the same distribution shift at every time step is an assumption or merely an artifact of the testbed.
- The number of clients (10 to 20) is too small for any conclusive claims in Federated Learning. Similarly, experiments with the MNIST and EMNIST datasets make the evaluation weak. The authors should attempt to find and experiment with some more realistic datasets, like the Portraits and Caltran datasets for the given setting.

---

> ### Author Response · Authors · 2025-06-26
> **Response to Reviewer 4V5m**
>
> We appreciate the reviewer’s efforts in reviewing our paper. Below, we respond to each of your concerns point by point. Revisions made in the manuscript based on your suggestions are highlighted in **pink**.
>
> ## Weakness 1: Parameter $ d $ and function $ f $
>
> These parameters have been introduced in the methodology part, as seen in Section 4.1 (below equation 4), where $d$ is a distance measure and $f$ is a mapping using parameter $\phi$ or $\psi$. We have also included these in Algorithms.
>
> ## Weakness 2: Algorithm 2 has parameter $\psi$  defined but w is used instead.
>
> Thanks for spotting the typo. We have fixed it in the revision.
>
> ## Weakness 3: The number of clients is only mentioned in the appendix.
>
> We have added it to the first paragraph of Section 5 in the revision.
>
> ## Weakness 4: The composition of the test set is not described. Does the test set follow the same label distribution as the training set? If so, why? In the experiments, how many domains were used during training and how many for testing? Consequently, it is unclear from the paper what the test accuracy really means.
>
> We clarify that this paper doesn't involve label shift between training and test, instead, it only considers the domain distribution shift over time, including shift between train and test sets, alongside class distribution heterogeneity across clients due to non-i.i.d. local data.
>
> Regarding the train-test domain split: as stated in Section 5.1, **"For all datasets, the last domain is viewed as the target domain."** This setup reflects the core setting of our work, which predicts performance on an unseen future distribution that evolves from the training domains. In line with this, we also emphasize in Section 1 one of our guiding research questions: *"How can we exploit the evolving patterns from training data (source domains) and deploy our model on the unseen future distribution (target domain)?"* Thus, the reported test accuracy measures model performance on this unseen distribution, capturing its generalization to future domains.
>
> ## Weakness 5: Whether all clients following the same distribution shift at every time step is an assumption or merely an artifact of the testbed?
>
> This is an assumption made for the purpose of theoretical analysis to simplify the representation and derivation. However, this is not a limitation of our method in practice. Notably, the **Portrait** and **Caltran** datasets are real-world benchmarks where clients do not follow strictly synchronized distribution shifts across domains. We include these datasets to demonstrate that our methods remain effective even when this assumption does not hold. In addition, we have also incorporated **Reddit** dataset in revision, which reflects our methods' performance in capturing language evolving patterns across years.
>
> ## Weakness 6: The number of clients is too small and experiments.
>
>  We follow the same number of clients setting as in this paper: Test-Time Robust Personalization for Federated Learning ICLR23, which is also using 20 clients. To further address this concern, we have scaled up the number of clients and included the corresponding results in Appendix E.5, which indicates that the number of clients would not influence the evaluation of model generalization ability.
>
> Regarding datasets, we already include the **Portraits** and **Caltran** datasets in multiple experimental settings to validate our methods under more realistic, complex scenarios. For other experiments such as those using MNIST-based datasets, our goal is to study specific influence of evolving distribution shifts to machine learning like evolving magnitude. They provide a controlled environment where the distribution shift is clear, quantifiable, and more challenging than the standard MNIST, making it well-suited for isolating and analyzing core behaviors of the proposed algorithms. We have also added the \textit{**Reddit**} dataset to the experiments to show the effectiveness of the method on text datasets.

---

> > ### Author Response · Authors · 2025-06-26
> >
> > ## Requested Changes 1 & 2: Simplify algorithms and add network diagram
> >
> > Thanks for your suggestions. We have followed your suggestions and updated the manuscript with a simplified version of the algorithms and a figure with a Neural Network.
> >
> > ## Requested Change 3: Add local-only baseline to show the impact of collaboration. This is especially important when the test set is not global and follows the data distribution of the training set.
> >
> > Note that, test set **does not** follow the data distribution of the training set. When $\text{dir} \rightarrow \infty$, the test set could be viewed as global since there is no heterogeneity between clients. To address your interest, we also added a local-only baseline to show the impact of collaboration in Appendix Section E.5 where it validates that the collaboration is useful and important in our problem.
> >
> > ## Requested Change 4: Why client models outperform server?
> >
> > The relative performance between client and server sides depends on the dataset characteristics and the nature of the distribution shifts. In the Portraits dataset, the evolving pattern is relatively subtle and does not significantly degrade performance. Changes in clothing or hairstyles over time have a minimal impact on the visual features relevant for gender classification. These features are largely consistent across individuals and easier to capture. As a result, the server-aggregated model is able to capture generalizable representations for gender detection, leading to stronger performance on the test data.

---

### Review · Reviewer_bAmS · 2025-06-12

**Summary Of Contributions:**

The paper introduces the setting of *federated learning under evolving distribution shifts*, where each client’s data distribution changes over time and the target distribution at test time is a future, unseen domain.

>1 It formalises the problem and shows that existing robust, personalised or test-time-adaptation FL frameworks do not address this scenario .

> 2 Two methods are proposed. **FedEvolve** uses a pair of representation functions and class prototypes to align consecutive domains, while **FedEvp** distils the same idea into a single model with an efficient personalisation step .

>3  The authors prove an upper bound (Theorem 4.1) that links the test error on the unseen domain to distances between learned representations across time, giving algorithmic guidance .

>4 Extensive experiments on synthetic (Circle) and four real datasets with image rotations, portraits and traffic scenes show large gains over fifteen baselines under varying client heterogeneity, number of domains and straggler ratios. Gains reach 15–30 pp accuracy on Rotated MNIST and Rotated EMNIST .

>5 An overhead study shows FedEvp matches baseline communication cost, whereas FedEvolve incurs roughly double the transmitted parameters .

**Audience:**

Yes

**Claims And Evidence:**

Yes

**Requested Changes:**

1.Add at least one non-image dataset with naturally evolving distributions, for example Reddit comment streams or MIMIC-IV time-stamped health data, to demonstrate modality generality.

2. Provide learning-curve plots and wall-clock time to convergence for FedEvolve, FedEvp and strong baselines to complement the parameter-count table.

3. Report memory footprint per client (parameters + prototypes) as the number of domains grows.

4. Clarify hyper-parameter tuning: list search ranges and budgets for every baseline and your methods to ensure fair comparison.

5. Discuss how prototype alignment interacts with privacy constraints. Could secure aggregation or differential privacy noise degrade the representation distances that the theory assumes?

6. Include a sensitivity study on the choice of distance metric (Euclidean vs cosine) and on prototype update schedules.


7. State all assumptions underlying Theorem 4.1 explicitly in the main text and comment on their realism in FL practice.

8. Release code and dataset splits upon publication to aid reproducibility.

Overall, the paper tackles an important, under-explored problem and proposes methods that are both effective and, in the FedEvp variant, efficient. Addressing the requested clarifications and broadening the empirical scope would further strengthen the submission.

**Strengths And Weaknesses:**

# Strengths

1. Distribution drift in real federated systems is common but rarely modelled; the work clearly motivates the gap with practical examples and literature survey .

2. Representation alignment with class prototypes is intuitive, differentiable and requires no access to future data. FedEvp pragmatically addresses FedEvolve’s overhead for resource-limited clients .

3. Theorem 4.1 and Lemma 4.2 connect representation distances to generalisation, explaining why aligning consecutive domains can bound future-domain risk .

4. Five datasets, three heterogeneity levels, ablations on number of domains, stragglers and personalisation detail yield a convincing empirical picture .

5. Appendix includes full pseudocode for both algorithms  and implementation details; ablation study clarifies design choices.



# Weaknesses
1. I might be wrong, but please double check this. The paper minimises Eq4.  which is already non-positive; driving it “down” pushes the log‐probability toward $-\infty$, i.e. it **maximises the classification error instead of minimising it**.  Standard prototypical or soft-nearest-neighbour training uses the *negative* log-likelihood, $-\log P(y\mid x)$.  The same sign error appears in the FedEvp implementation of $\ell_f$  and in Algorithm 1 lines 14–17 .

2.  The proof in Appendix B relies on the triangle inequality for $d_{\mathrm{JS}}$ , yet the Jensen–Shannon divergence satisfies the triangle inequality **only after taking the square-root**.  If $d_{\mathrm{JS}}$ is defined without the root (as in the text and experiments), Steps (2)–(3) of the proof do not hold.

3. The bounds rely on bounded losses and Jensen–Shannon distances between consecutive domains but do not analyse optimisation error or stochastic variance introduced by FL; practical implications are not quantified.

4. All real tasks are image classification. It is unclear whether the approach generalises to text, tabular health records or time-series modalities typical in FL.

5.  Doubling the encoder increases wall-clock time by over 100 percent in some settings , yet no convergence-time comparison is provided.

6. Storing class prototypes for every past domain may become costly when the number of domains or classes is large; memory complexity is not analysed.

7. Prototype sharing could leak information. The paper does not discuss privacy amplification, gradient-based attacks or interactions with differential privacy.

---

> ### Author Response · Authors · 2025-06-26
> **Response to Reviewer bAmS**
>
> We sincerely thank the reviewer for the thoughtful and constructive feedback. Below, we respond to each concern point by point. Revisions made in the manuscript based on your suggestions are highlighted in **blue**.
>
> ## Weakness 1: Sign error
>
> Thanks for reminding our typos. We have fixed them in the updated manuscript.
>
> ## Weakness 2: The proof in Appendix B
>
> Thanks for spotting the typo. In the proof of Thm 4.1, we indeed mean the Jensen-Shannon distance (which is the square root of $\sqrt{d_{JS}}$) satisfies the triangle inequality because the Lemma we use (Lemma 1 in Wang et al., ) is on the square root of $d_{JS}$ and we made a typo. We modified the typo in the proof in Appendix B and ensured Theorem 4.1 aligns with the proof. Note that we still use $d_{JS}$ as the Jensen-Shannon divergence in the main paper and add square roots to it. But we define $D_{JS} = \sqrt{d_{JS}}$ as the Jensen-Shannon distance metric in the proof of Appendix. We highlight the modified parts in blue text. The correction of this typo does not influence any interpretation of Theorem 4.1.
>
> ## Weakness 3: Practical implications
>
> The bounds in our theorem 4.1 upper bound the classification error into the summation of 4 terms, where term 2 and 4 demonstrate the additional errors brought by evolving domains and term 3 demonstrates the heterogeneity brought by FL. The theoretical results are used to justify the design of algorithms, while the comprehensive experimental results quantify the practical usages.
>
> ## Weakness 4 and Requested Changes 1: Non-image datasets
>
> In our experiments, we use the dataset Circle, which is a non-image dataset, to benchmark our methods. To further address this concern, we have included the **Reddit** streams dataset and use a text classification task to evaluate the methods. Due to time limitations, we did not run experiments for all baselines but selected those with stronger performance compared to others. The results are provided in Appendix E.2, where our methods still outperform baselines on text datasets.
>
> ## Weakness 5 and Requested Changes 2: Convergence
>
> Following your suggestion, we have added learning-curve plots and wall-clock time in Figure 4.
>
> ## Weakness 6 and Requested Changes 3: Memory complexity
>
> We clarify that our method does not store class prototypes for every past domain. As shown in Algorithms 1 and 2 (Algorithms 3 and 4 in the revision), at each iteration, we sample data from two consecutive domains and compute prototypes on-the-fly from the sampled data. Thus, at any given time, we only store prototypes for a single domain.
>
> Using FP32 precision, each prototype is a vector with at most 2048 dimensions. Even in a scenario with 1,000 classes, the total memory required is approximately 8 MB, which is practical for modern federated devices.
>
> ## Weakness 7: Prototype sharing could leak information
>
> We clarify that prototypes are not shared between clients. All prototypes are computed and used locally on each client during training and evaluation. Therefore, no prototype information is transmitted or exposed across the network. Instead, we follow the FL routine that only shares model weights between the server and clients.
>
> ## Requested Changes 4: hyper-parameter
>
> Our hyperparameter tuning details are provided in Appendix D and in the table titled *Training Details for Datasets*. Following your suggestion, we have extended these sections to include the search strategy and range for baselines.
>
> ## Requested Changes 5: Privacy constraints
>
> Our methods do not transmit prototypes across clients or to the server; all prototype computations are based on model hidden layers before the classifier layers and updates are performed locally. Only the models/gradients would be shared between the server and clients. As a result, prototype alignment does not introduce additional privacy risks under the standard FL setting. However, if differential privacy is applied, the added noise may affect the quality of learned representations and weaken prototype alignment. Moreover, as FedEvolve involves more parameters (two mappings), achieving a given privacy budget may require injecting more noise and impact utility. Investigating this interaction is an important direction for our future work.
>
> ## Requested Changes 6: Sensitivity studies
>
> We have included a new table in Appendix E.3 presenting a sensitivity analysis on the choice of distance metrics, comparing cosine distance and L2 distance. Regarding prototype update schedules, our method computes prototypes at every iteration using sampled data from the current domain. Therefore, it does not rely on a fixed update frequency or any hyperparameter to control prototype updates.
>
> ## Requested Changes 7: Assumptions
>
> We have corrected the typo in Theorem 4.1.
>
> ## Requested Changes 8: Reproducibility
>
> The algorithm implementation is given in https://anonymous.4open.science/r/3D68/. The complete dataset code will be released after organizing the code.

---

### Decision · Action_Editor_8K1w · 2025-07-25

**Recommendation:** Reject

**Audience:**

Yes

**Audience Explanation:**

The general topic is of interest, but I will caveat this with saying that the actual experimental setting reduced the reviewers' belief that the results of the paper would be relevant to TMLR audience (e.g. due to unrealistic experimental settings, analysis of settings where it is unclear whether clients would want to cooperate at all, and concerns about scalability).

**Claims And Evidence:**

No

**Claims Explanation:**

Reviewers were generally concerned that the significance of the empirical results in the work were quite small, especially compared to the complexity of the proposed algorithm. Multiple reviewers mentioned unexplained, unsupported, or misleading claims and discussions. Moreover, a general critique shared across reviewers is that the empirical results even in perhaps unrealistic settings with major amounts of distribution shift were still not that significant (again, with respect to the complexity of the algorithm in question).

I think that the reviewers have done a great job identifying points of confusion with respect to the main claims made in the paper, and that incorporation of these into improving the demonstration of these claims would greatly improve the work.

**Resubmission Of Major Revision:**

The authors may consider submitting a major revision at a later time.